Spatial patterns in the contribution of biotic and abiotic factors to the population dynamics of three freshwater fish species

Chevalier Mathieu mathieu.chevalier@ifremer.fr 1
Tedesco Pablo 2
Grenouillet Gael 2
1 Centre de Bretagne, DYNECO, Laboratoire d’Ecologie Benthique Côtière (LEBCO), IFREMER , Plouzané , France
2 Laboratoire Évolution & Diversité Biologique (EDB), CNRS, Université de Toulouse , Toulouse , France
Baird Donald
Electronic publication date: 2022 Feb 23
Publication date: 2022
Volume: 10
Electronic Location ID: e12857
Received 2021 Jul 15; Accepted 2022 Jan 9
Copyright: ©2022 Chevalier et al.
Copyright year: 2022
Copyright holder: Chevalier et al.
License: This is an open access article distributed under the terms of the Creative Commons Attribution License, which permits unrestricted use, distribution, reproduction and adaptation in any medium and for any purpose provided that it is properly attributed. For attribution, the original author(s), title, publication source (PeerJ) and either DOI or URL of the article must be cited.
License URL: https://creativecommons.org/licenses/by/4.0/

Keywords: Water temperature, Density-dependence, Spatial variation, Size classes, Population dynamics, Length-frequency histograms, Abundant-center hypothesis, Range shifts

Funding: CEBA ANR-10-LABX-0025 TULIP ANR-10-LABX-41 The EDB lab was supported by ‘Investissement d’Avenir’ grants (CEBA, ref. ANR-10-LABX-0025; TULIP, ref. ANR-10-LABX-41). The funders had no role in study design, data collection and analysis, decision to publish, or preparation of the manuscript.

==============================
Background

Population dynamics are driven by a number of biotic (e.g., density-dependence) and abiotic (e.g., climate) factors whose contribution can greatly vary across study systems (i.e., populations). Yet, the extent to which the contribution of these factors varies across populations and between species and whether spatial patterns can be identified has received little attention.

Methods

Here, we used a long-term (1982–2011), broad scale (182 sites distributed across metropolitan France) dataset to study spatial patterns in the population’s dynamics of three freshwater fish species presenting contrasted life-histories and patterns of elevation range shifts in recent decades. We used a hierarchical Bayesian approach together with an elasticity analysis to estimate the relative contribution of a set of biotic (e.g., strength of density dependence, recruitment rate) and abiotic (mean and variability of water temperature) factors affecting the site-specific dynamic of two different size classes (0+ and >0+ individuals) for the three species. We then tested whether the local contribution of each factor presented evidence for biogeographical patterns by confronting two non-mutually exclusive hypotheses: the “range-shift” hypothesis that predicts a gradient along elevation or latitude and the “abundant-center” hypothesis that predicts a gradient from the center to the edge of the species’ distributional range.

Results

Despite contrasted life-histories, the three species displayed similar large-scale patterns in population dynamics with a much stronger contribution of biotic factors over abiotic ones. Yet, the contribution of the different factors strongly varied within distributional ranges and followed distinct spatial patterns. Indeed, while abiotic factors mostly varied along elevation, biotic factors—which disproportionately contributed to population dynamics—varied along both elevation and latitude.

Conclusions

Overall while our results provide stronger support for the range-shift hypothesis, they also highlight the dual effect of distinct factors on spatial patterns in population dynamics and can explain the overall difficulty to find general evidence for geographic gradients in natural populations. We propose that considering the separate contribution of the factors affecting population dynamics could help better understand the drivers of abundance-distribution patterns.

Introduction

Population dynamics have been related to several factors that can be classified as intrinsic (i.e., biotic) or extrinsic (i.e., abiotic) (Cappuccino & Price, 1995). Whether populations are mostly influenced by one or the other type of factor has been a matter of debate (Andrewartha & Birch, 1954; Nicholson, 1957) and has recently regain interest owing to the need to improve our knowledge regarding the influence of climate change on population and species extinction risk (Bellard et al., 2012). Although there is now clear evidence that both factors can influence wild populations (Bjørnstad & Grenfell, 2001), we still have a poor understanding of their relative influence across the distributional range of species (Frederiksen, Harris & Wanless, 2005; Guo et al., 2005; Pearce-Higgins et al., 2015; Dallas, Decker & Hastings, 2017; Pironon et al., 2017).

Beyond data limitations (i.e., the need of long-term time series for multiple populations), the complexity underlying population dynamics may explain this lack of knowledge (Benton, Plaistow & Coulson, 2006). For instance, in stage-structured populations, individuals in different states can be differentially influenced by climatic conditions (Coulson et al., 2001; Ibáñez et al., 2015) while contributing differently to the overall dynamic of the population (Radchuk, Turlure & Schtickzelle, 2013). At the same time, climate can have various influences on populations depending on their position within the distributional range of species (Curnutt, Pimm & Maurer, 1996; Sæther et al., 2008; Pearce-Higgins et al., 2015). Some studies have attempted to identify the large-scale drivers of population dynamics by studying the extent to which spatially distant populations tend to vary in synchrony over time (Liebhold, Koenig & Bjørnstad, 2004). While in some cases, their findings support the view of a common climatic driver (e.g., temperature) affecting populations abundance similarly, in most cases, a considerable amount of variance remains unexplained (Chevalier et al., 2015), suggesting that population dynamics and their associated drivers can greatly vary over space (Sæther et al., 2008; Grøtan et al., 2009; Roy, McIntire & Cumming, 2016).

Spatial differences in population dynamics can vary according to a number of factors including species range limits (Williams, Ives & Applegate, 2003), the abiotic environment (Sæther et al., 2008), resource availability (Wang et al., 2008), latitude (Sæther et al., 2008; Pearce-Higgins et al., 2015) or elevation (Dostálek, Rokaya & Münzbergová, 2018). Two main hypotheses have been advanced to explain these spatial variations. Based on the niche concept (Hutchinson, 1957; Peterson et al., 2011), the first hypothesis predicts a negative relationship between population abundance and the distance to the geographic range center (Brown, 1984). This abundant-center hypothesis (also called core–periphery hypothesis; Pironon et al., 2017) has a long-standing history in ecology and assumes that environmental conditions become harsher towards the limits of species ranges, leading to geographic patterns in the demographic performance, the strength of density-dependence or the genetic variability of populations (Sagarin, Gaines & Gaylord, 2006). For instance, a main expectation is that core populations should be rather regulated by density-dependent processes because highly productive areas tend to be monopolized by individuals with high competitive abilities, whereas peripheral populations are rather expected to be regulated by abiotic factors (Pironon et al., 2017; Santini et al., 2019). However, recent studies found contrasting empirical support for the abundant-center hypothesis casting doubts about its generality (Dallas, Decker & Hastings, 2017; Santini et al., 2019). The second hypothesis—the range-shift hypothesis—is based on the evidence that species are moving poleward or upward to track suitable climatic conditions (Parmesan & Yohe, 2003; Pecl et al., 2017). These range shifts imply that populations located at the periphery of the range can display various behaviors depending on whether they are located at the trailing (i.e., low latitude or elevation) edge where extinction processes are at play (Kuussaari et al., 2009) or at the leading (i.e., high latitude or elevation) edge where colonization processes should be more prevalent (Engler et al., 2009). Different spatial patterns are expected under this hypothesis. For instance, while trailing and leading-edge populations should be both weakly regulated by density-dependent processes owing to environmental disequilibrium, populations located at the leading edge are expected to be positively affected by climate change, whereas the opposite is expected for populations located at the trailing edge (Mills et al., 2017). Whether spatial variations in population dynamics are best explained by the abundant-center or the range-shift hypothesis remains unexplored to date.

In this study, we used an extensive database containing information on population abundances and individual sizes from different sites covering France, to study the spatial pattern in the contribution of biotic and abiotic (temperature-related) factors to the population dynamics of three freshwater fish species presenting contrasted life-histories and patterns of range-shifts (see below). We used length frequency histograms (Cattanéo & Lamouroux, 2002; Bergerot & Cattanéo, 2017) to separate individuals into two size-classes corresponding to young-of-the-year (0+) and older individuals (>0+). This allowed us to study spatial variations regarding the strength of density dependence between >0+ individuals (owing to competition for resources), the productivity rate of >0+ individuals (i.e., an equivalent of the population growth rate but tailored to this particular size-class), the apparent recruitment rate of 0+ individuals (which depends on the density of >0+ individuals) and the apparent survival rate of 0+ individuals (Grenouillet et al., 2001). Regarding abiotic factors, we focused on the effect of temperature, a factor known to be a strong determinant of the abundance of 0+ individuals (Grenouillet et al., 2001) and year-class strength (i.e., the number of larvae hatched in a given year) and which is therefore classically considered as the most important factor in fish ecology (Mills & Mann, 1985). However, while previous studies mostly focused on changes in average temperature, here we also considered the effect of changes in temperature variability; a component predicted to be strongly affected by climate change (Lawson et al., 2015). We used Bayesian inference to estimate model parameters affecting each species population dynamics and elasticity analysis to highlight the relative contribution of biotic and temperature-related factors to the population dynamics of the three species (Koons et al., 2015). From elasticity measures, we then asked the following questions: (i) is the contribution of biotic and temperature-related factors similar across species presenting different life-histories, (ii) can we identify a spatial pattern in the contribution of the different factors and if so (iii) which of the range-shift or the abundant-center hypothesis best explains the observed pattern? The two latter questions were tackled through a model selection procedure testing differences between a null model (assuming no spatial pattern), a model that includes the distance to the geographic center as a covariate (abundant-center hypothesis) and two models that either included latitude or elevation as covariates (range-shift hypothesis). To the best of our knowledge, this study is the first using data from a monitoring program together with a modelling framework integrating elasticity analyses to derive inferences about the drivers of spatial variations in population dynamics, while accounting for stage-specific dynamics.

Materials and Methods

Datasets

Studied species

We considered three species presenting different life-history strategies and patterns of elevational range shifts: the barbel (Barbus barbus), the roach (Rutilus rutilus) and the chub (Squalius cephalus) (Table 1). Following the three demographic strategies proposed by Winemiller (1992), the barbel is an ‘equilibrium’ strategist characterized by a long lifespan, a low fecundity and a large body size (Kottelat & Freyhof, 2007; Froese & Pauly, 2021), the roach is an ‘opportunistic’ strategist characterized by a small size and a low fecundity whereas the chub has the opposite characteristics and can be considered a ‘periodic’ strategist. The opportunistic strategy should maximize the colonizing capability of species in stochastic environments with frequent changes at small temporal and spatial scales. Alternatively, a periodic strategy is favored in environments with large scale cyclic variations (e.g., seasonal environment), whereas an equilibrium strategy is favored in environments with low temporal variation in habitat quality and strong biotic interactions (Winemiller, 1992).

Table 1 Life-history attributes and range shifting patterns along the elevational gradient for the three freshwater fish species.

Values were taken from various sources (Froese & Pauly, 2021; Kottelat & Freyhof, 2007; Comte & Grenouillet, 2013).

	Roach	Chub	Barbel	
Fecundity	50,000	125,000	10,000	
Body length (mm)	275	400	500	
Lifespan (years)	14	16	20	
Critical thermal maximum (°C)	39.00	38.00	32.00	
Range size (km2)	19,522,376	7,126,749	2,782,586	
Shift trailing edge (m/yr)	0.022	0.054	0.115	
Shift leading edge (m/yr)	−0.873	1.542	−4.311	
Shift centroid (m/yr)	0.266	0.209	−0.125	
Strategy	Opportunistic	Periodic	Equilibrium	

These differences are expected to entail variations in the direction of the effect and the relative contribution of biotic and temperature-related factors to the population dynamics of the three considered species. For instance, while the abundance of all three species is expected to be positively affected by water temperature (Mills & Mann, 1985; Grenouillet et al., 2001; Piffady et al., 2010), we expect the barbel to show a stronger regulation by density, particularly for populations located at the center of the range, while the two other species are expected to be rather regulated by temperature-related factors with an increasing negative contribution as the distance to the geographic center increases (Williams, Ives & Applegate, 2003). Similarly, we expect temperature variability to have a larger positive contribution on roach abundances (opportunistic strategy) than on the abundance of the two other species where average temperature could have a stronger (and positive) influence (Winemiller & Rose, 1992).

The three species also present different patterns of elevational range shifts in the last decades (Comte & Grenouillet, 2013), with different responses observed along the elevational gradient (Table 1). For instance, the leading edges (high-elevation populations) of barbel and roach have shifted downward, whereas an upward shift was recorded for chub (Table 1). Thus, a positive influence of temperature is expected for high-elevation populations of chub while the opposite is expected for the two other species. The trailing edges (low-elevation populations) of the three species have been observed to shift upward but at a different pace. The larger shift observed for barbel (Table 1) could be associated with a stronger and negative contribution of temperature for low-elevation populations.

Species data

Fish population abundances and individual sizes were extracted from the freshwater fish monitoring database of the French Biodiversity Office (OFB, http://www.image.eaufrance.fr). We selected 182 sites where data was collected between 1982 and 2011 (3,143 sampling operations) by electrofishing. Streams were sampled by wading, during periods of low flow (from May to October), after the reproduction time, using a point sampling strategy covering the different habitats (e.g., pools, riffles, submerged vegetation) observed over the study sites (Poulet, Beaulaton & Dembski, 2011). Fish were identified to species level, measured for total body length, counted, and released to the river. For the three species, we selected time series that were composed of at least 15 years of data during which the sampling protocol remained the same and contained at least 50% of non-null captures at the population level (i.e., 0+ and >0+ individuals confounded). This selection was made to reduce the number of zeroes while keeping times series of sufficient length to allow for an appropriate estimation of the temporal dynamic of populations. We further discarded time series with more than three consecutive years missing to ensure that the loss of information in population change during the missing years is minimized (Engen et al., 2005). Missing values were ignored during the modelling process. This selection process ensures reliable model inference and left us with 71 (mean length = 17.09 years), 175 (mean length = 17.24 years) and 152 (mean length = 17.26 years) time series for barbel, chub and roach, respectively. In total, 326,234 individuals were collected. Further details about abundance and size data are provided in Appendix S1, Figs. S1 and S2.

Temperature data

Daily air temperatures from 1982 to 2011 were provided by Météo France and extracted from the high resolution (8 km by 8 km grid) SAFRAN atmospheric analysis over France (Le Moigne, 2002). Daily water temperature data measured from 2009 to 2012 at 135 sites located throughout France were provided by the French Biodiversity Office (https://ofb.gouv.fr/). From these two datasets, we used a random forest algorithm where we modelled model water temperature as a function of three covariates: air temperature, month and elevation. The model showed a very good performance and was therefore used to predict daily water temperatures for all sampling sites over the course of the study period. For further details see Chevalier et al. (2018). From daily predictions, we calculated the annual mean and intra-annual variability of water temperature between consecutive sampling occasions at each site and used these temperature variables as covariates in the population dynamic models. For each species, a summary of both variables is provided in Appendix S1; Fig. S3. These variables were transformed to z-scores before model fitting to improve model convergence.

Geographic range data

The above-mentioned abundance data do not encompass the full geographic range of the species, potentially leading to niche truncation issues and biased location of geographic range centers (Knouft, 2018; Soberón, Townsend Peterson & Osorio-Olvera, 2018; Dallas, Pironon & Santini, 2020). To obtain an unbiased estimate of the location of range centers, we used IUCN range maps (https://www.iucnredlist.org/resources/spatial-data-download). Specifically, for each species, we computed its geographic range center as the center of IUCN polygons (based on geographic coordinates) using the gCentroid function of the package rgeos (Bivand & Rundel, 2018).

Statistical analyses

The modelling workflow (Fig. 1) can be decomposed in four steps, where (1) abundance data are determined for each size class based on individuals’ length measured at each sampling operation, (2) the dynamic of the two size-classes is modelled using Bayesian inference, (3) an elasticity analysis is conducted to estimate the contribution of biotic and temperature-related factors on the dynamic of each size class, and (4) a model selection procedure is conducted to investigate the spatial pattern in the contribution of the different parameters and determine whether this pattern rather corresponds to the range-shift or the abundant-center hypothesis.

Figure 1 Analyses workflow.

Analyses workflow describing (first column) how length-frequency histograms were used to discriminate the two size-classes and obtain abundance data for each size-class for all sampling operations and (second column) how abundance data were used to extract population dynamics parameters, perform the elasticity analysis and investigate the spatial patterns in the contribution of biotic and abiotic factors to the population dynamics of the three species.

Discriminating 0+ and >0+ individuals

For each species, we used the length-frequency histograms of each sampling event to separate individuals into two size classes (Bergerot & Cattanéo, 2017) using Gaussian components (McLachlan & Peel, 2004). This algorithm assumes that length data can be described by a mixture of two normal distributions which correspond in our case to the length frequency distributions of 0+ and >0+ individuals. The parameters of the two distributions were estimated using an expectation–maximization algorithm and the limit between the two size classes was fixed at the length where the two distributions crossed. However, because the algorithm performed poorly when the separation between the two size classes was not evident, e.g., when there were few individuals in each group, this procedure cannot be routinely applied to discriminate 0+ and >0+ individuals for each sampling operation. Therefore a few additional steps had to be considered (see Fig. 1; left column). For each species, we first selected 20 length-frequency histograms for which the discrimination between the two size classes was visually clear and assigned each individual to the 0+ or >0+ group based on the estimated length limit (Fig. 1). To discriminate 0+ from >0+ individuals for the remaining sampling events, we used a random forest approach (Liaw & Wiener, 2002) where individual status (0+ or >0+) was modeled as a function of individual size, individual numbers (to account for potential effects of density dependence) and annual cumulative degree-days where the water temperature was above 12 °C (i.e., the temperature below which growth is assumed inhibited; Nunn et al., 2003). The model was calibrated using the 20 sampling events for which individual status was assumed unbiased. The predictive performance of our model was tested by running a split-sample cross-validation procedure 100 times. This procedure revealed a very good model performance in predicting individual status, as measured with the Cohen’s kappa coefficient (κ > 0.99 for the three species; Cohen, 1960). We therefore used this model to predict individual’s status for the remaining sampling events. For each species, individuals in each size class were summed for each sampling event to obtain abundance time series (Appendix S1; Fig. S2).

Population dynamics model

The abundance of individuals in each size class was modelled on the log-scale using two normal distributions (one for each size class):

Xi,t0+∼Normalλi,t0+,σ0+

Xi,t>0+∼Normalλi,t>0+,σ>0+

where Xi,t is the log-abundance of individuals in each size class (0+ and >0+) at site i and time t, λi,t is the expected log-abundance and σ is the associated process error variance. The means of the two distributions (i.e., λi,t0+ and λi,t>0+) were modeled with different functional forms to account for variation in the underlying demographic process (see Grenouillet et al., 2001 for a similar approach). Specifically, the dynamic of 0+ individuals was modelled as: λi,t0+=αi0++βi0+×Xi,t>0+ logSi,t+ ∑j=1Jγi,j0+×Uj,i,t+ logSi,t

where αi0+ is a site varying intercept, βi0+ is a density-dependent parameter reflecting the dependency of 0+ individuals to the abundance of >0+ individuals at time t and can be interpreted as the apparent recruitment rate (since sampling takes place after reproduction) and Si,t is the sampling area (offset term). Note that the recruitment rate is only ‘apparent’ because (1) the >0+ size-class includes both spawners and non-spawners (reported age at maturity for females is 3–4 years for the chub, 2–3 years for the roach and 5 years and more for the barbel; Keith et al., 2011; Keith et al., 2020) and thus also accounts for the effect of competition with non-spawners and (2) inferences are based directly on the abundance of recruits (i.e., 0+ individuals that successfully hatched), hence not accounting for variation in the per capita reproductive investment (i.e., the number of eggs laid by a given individual). A more accurate estimation of the recruitment rate could be achieved using egg data together with data on spawner biomass/abundance. The parameters γi,j0+ are regression coefficients applied to the array Uj,i,t which contains the raw and the squared values of the mean and the variability of water temperatures at site i and time t. Thus, γi,j0+ is a vector of coefficients representing the linear and the quadratic effect of the mean and the variability of water temperature on 0+ abundance at each site.

The dynamic of >0+ individuals was represented using a modified version of the stochastic Gompertz model of population growth as: λi,t>0+=αi>0++Xi,t−1>0++βi>0+×Xi,t−1>0+ logSi,t−1+δi>0+×Xi,t−10+ logSi,t−1+∑j=1Jγi,j>0+×Uj,i,t+logSi,tSi,t−1

where αi>0+ is a site varying intercept representing the intra-class productivity rate (an analog to the population growth rate with values above one indicating positive productivity rates), βi>0+ is a density-dependent parameter representing the competition between >0+ individuals for access to resources and δi>0+ represents the transition probability between the two size classes (i.e., the apparent survival rate of 0+ individuals). Similar to the recruitment rate, we note here that the survival rate is only ‘apparent’ because the >0+ size-class includes individuals in different ages. This survival rate thus also accounts for the survival probability of all other size-classes. The parameters γi,j>0+ are regression coefficients representing the linear and the quadratic effects of the two temperature variables on >0+ abundance.

Quadratic effects were included in both dynamics to account for potential bell-shaped response curves along the temperature gradient (Austin, 1999). The model was fitted to each species separately, and included random site effects for all population dynamic parameters, ultimately making it possible to analyze spatial patterns in the contribution of biotic and temperature-related factors to the dynamic of each size-class.

Parameter estimation and model goodness of fit

The model was fitted to each species using Bayesian inference and weakly informative priors. Site-specific parameters (αi0+,βi0+,γi,j0+,αi>0+, βi>0+, δi>0+, γi,j>0+) were assumed to follow normal distributions with a vector of means µ{μα0+, μβ0+, μγj0+, μα>0+, μβ>0+, μγj>0+, μδ>0+} and of standard deviations σ{ σα0+, σβ0+, σγj0+, σα>0+, σβ>0+, σγj>0+, σδ>0+}. The vector µrepresents the average value of the parameters across all sampling sites whereas the vector σ represents departures from the mean and therefore the spatial variability in parameter values. We used normal distributions with mean zero and standard deviations of 10 as priors for all µ. For σ, σ0+ and σ>0+, we used half-Cauchy distributions (Gelman, 2006). For each species, we generated three chains of length 11,000 with initial values in different regions of parameter space and discarded the first 1,000 iterations as burn-in. Chains were sampled every 10 iterations. Convergence was visually assessed and confirmed using the Gelman and Rubin statistic with a threshold value of 1.1 (Gelman & Rubin, 1992). Highest Posterior Density (HPD) intervals were used as 95% credible intervals. For each parameter, differences between species were assessed by computing the proportional overlap between the two posterior distributions. A low overlap (threshold set to 5% meaning that only 5% of MCMC samples were common between the two distributions) was taken as evidence that estimated parameters were different between species.

We used posterior predictive checks (Gelman, Meng & Stern, 1996) to assess the goodness of fit of our model for the three species. Specifically, we used χ2 discrepancy metrics to compute the posterior predictive p-value, which quantifies the extent to which the proportion of samples in which the distance of observed data to the model is greater than the distance of replicated data to the model. Values close to 0.5 indicate a good model fit, whereas values close to 0 or 1 indicate lack of fit. Bayesian p-values were calculated regarding the log-abundance of both 0+ and >0+ individuals. We fitted the models using JAGS 4.3.0 (Plummer, 2003), run through the R environment (R Core Team, 2019) using the packages R2jags (Su & Yajima, 2013) and rjags (Plummer, 2014). The JAGS code is available in Appendix S2.

Elasticity analyses

We applied an elasticity analysis to species model outputs in order to highlight the relative contribution of biotic (αi0+,βi0+,βi>0+ and δi>0+) and temperature-related (γi,j0+ and γi,j>0+) factors to the population dynamics of the three species following the framework developed by Koons et al. (2015). Specifically, for each species, we used the median of the posterior distribution of parameters obtained from the fitted model to project the log-abundance of both size-classes at each site over the study period (Haridas, Tuljapurkar & Coulson, 2009). This was done by iteratively updating the parameter λi,t for both 0+ and >0+ individuals using observed predictor values for the parameter for which elasticity needs to be calculated but average predictor values for the other parameters. We measured the contribution of biotic and temperature-related factors separately for each size-class by comparing the log-abundance computed using parameter values predicted by the model (θori) to the log-abundance computed by changing each parameter value, one at a time by 10% (θper; both the linear and the quadratic terms were changed for temperature-related factors). Specifically, elasticities were computed numerically as: eω,i,t=θper,i,t−θori,i,tθori,i,t×1δ

where θ is the response parameter (i.e., log-abundance of the considered size-class original and perturbed) at site i and time t, ω is the parameter of interest and δ is the proportional change in ω (i.e., 10%). The mean of all eω,i,t, therefore represents the estimated elasticity of parameter ω at the species level (eω,sp). For each parameter, differences between species were tested using Wilcoxon signed-rank tests with p-values adjusted for multiple comparisons using the Bonferroni correction.

Spatial patterns in the contribution of biotic and temperature-related factors

In order to establish how the contribution of biotic and temperature-related factors varied spatially, we built four different linear models: an intercept-only model (null model against which the other models are compared to), a model with elevation as covariate (elevation range-shift model), a model with latitude as covariate (latitude range-shift model) and a model with the Haversine distance (i.e., Euclidean distance accounting for the curvature of the Earth) to the geographic range center (abundant-center hypothesis model; Soberón, Townsend Peterson & Osorio-Olvera, 2018) as covariate. The three models with covariates included quadratic terms to account for non-linear effects. Models were run for the eight parameters for which elasticity was computed and compared using AIC (Burnham & Anderson, 2002). For each species, further details for the three covariates is provided in Appendix S1; Fig. S4.

Results

The Bayesian population dynamics models converged well for all three species (potential scale reduction factor less than 1.1 for all parameters). The posterior predictive checks revealed very good model fits either for 0+ or >0+ log-abundance with Bayesian p-values close to 0.5 in all cases.

Global patterns

Despite different life-history strategies, the large-scale ecological determinants of the population dynamics of the three species were similar, though the magnitude of effects varied across species (Fig. 2). For all species, the productivity rate of >0+ individuals suggested that populations had the potential to grow from low densities (α>0+>1; Fig. 2), particularly for chub (HPD95% = [1.42–1.71]). Furthermore, all species presented a positive recruitment rate (β0+ > 0) as well as evidence for a regulation of the dynamic of >0+individuals by density (β>0+ < 0) potentially owing to competition for resources (Fig. 2). However, barbel presented a tendency for a lower recruitment rate (HPD95% [1.03–2.18]) and a stronger regulation by density (HPD95% = [−4.11 to −5.19]) than the two other species. Chub (HPD95% [0.24–0.60]) and particularly roach (HPD95% [0.56–1.11]) also presented evidence for a positive survival rate of 0+ individuals (δ>0+>0) whereas the effect was more uncertain for barbel (HPD95% [−0.28–0.46]; Fig. 2). All species presented a tendency for a linear (all quadratic terms have their HPD95% overlapping zero) and rather positive effect of temperature-related factors on the dynamic of both 0+ and >0+ individuals, although with some uncertainty (Fig. 2). For instance, barbel (HPD95% [0.14–0.48]) and to a lower extent chub (HPD95% [0.04–0.27]) presented evidence for a positive effect of temperature variability on 0+ abundances whereas for roach we rather found a positive effect of average temperature on the abundance of >0+ individuals (HPD95% [0.02–0.18]).

Figure 2 Posterior summary of model parameters.

Posterior summary of model parameters (small-size panels with dots representing the median of the posterior distribution and the vertical line representing the associated 95% credible interval; horizontal dashed line point to the zero value) and corresponding relationships (large-size panels with lines representing the median of the posterior distribution and shaded areas representing the associated 95% credible interval) for the three species and the two size classes. The first column (except the first panel) highlights the effect of biotic (β0+) and abiotic (γ0+) factors on the abundance of 0+ individuals whereas the second column (and the first panel of the first column) highlights the effect of biotic (α<0+, β<0+, δ<0+) and abiotic (γ<0+) factors on the growth rate (log) of >0+ individuals. Differences between model parameters were tested by computing the proportional overlap between posterior distributions (ns > 5% overlap, ∗ < 5% overlap, ** <0.01% overlap, *** <0.001% overlap).

Relative contribution of biotic and temperature-related factors to population dynamics

Globally, the elasticity analysis indicated that the log-abundance of the three species was most sensitive to changes in biotic factors acting on the dynamics of >0+ individuals than on the dynamics of 0+ individuals (Fig. 3). For instance, a 10% change in the strength of density dependence (β>0+) would, on average, induce a 10.1% decrease in the log-abundance of >0+ individuals while a 10% change in the recruitment rate of 0+ individuals (β0+) would only entail a 0.9% increase in the log-abundance of 0+ individuals. Temperature-related factors only had a marginal contribution with the cumulative effect over both size classes only inducing a 0.7% change in log-population abundance on average. Yet, a number of populations presented elasticity values close to 0.4 (meaning a 4% change), indicating that temperature-related factors can have important local effects on the dynamic of both size-classes (Fig. 3).

Figure 3 Boxplots representing site-specific elasticities on log-abundances of 0+ or >0+ individuals to a 10% proportional change in the value of coefficients associated to biotic and abiotic factors for the three species with negative values pointing to a decrease in population abundance.

Dots represent the average elasticity for a given site. The horizontal dashed line points to the zero value. Grey zones show coefficients affecting the dynamic of 0+ individuals whereas white zones show coefficients affecting the dynamic of >0+ individuals. For temperature-related factors, both the linear and the quadratic coefficients were changed when calculating elasticities in order to obtain one elasticity value for each variable. Note that a boxplot overlapping zero does not mean that there is no effect but that the positive effect of the variable on some populations is counterbalanced by the negative effect of that variable on other populations.

Overall, chub was the species presenting the largest elasticities (mean = 0.34, SD = 0.47) followed by roach (mean = 0.32, SD = 0.25) and then barbel (mean = 0.16, SD = 0.19). This tendency was congruent and statistically significant (Wilcoxon test p-value < 0.05) for most biotic parameters, except the recruitment rate, were roach presented the largest elasticity (Fig. 3). Regarding temperature-related factors, although most comparisons were statistically significant, the differences were anecdotal relative to biotic factors.

Spatial patterns in the contribution of biotic and temperature-related factors to population dynamics

Despite evidence for large-scale determinants of population dynamics, the contribution of biotic and temperature-related factors strongly varied depending on the spatial location of populations (Fig. 4). For most factors we found that these spatial variations were mostly related to elevation and less so to latitude or the distance to the geographic range center (Table 2).

Figure 4 Species range (grey polygons) and associated geographic centers (black square) along with spatial variation in the contribution of biotic and abiotic factors to the population dynamic of the three species across metropolitan France.

Species are in lines while coefficients are in columns. Each point represents a population with colors corresponding to the estimated elasticity. The background surface represents elevation. Maps on grey background show coefficients affecting the dynamic of 0+ individuals while the other maps show coefficients affecting the dynamic of >0+ individuals. Note that the color scale is different in each panel.

Table 2 Results of model selection performed on the different populations dynamic parameters for the three species.

For coefficients and R2 values of the most supported model; see Fig. 5.

Species	Factor name	Parameters	AIC null model	AIC elevation model	AIC distance model	AIC latitude model	Most supported model	
Barbel	Average temperature	μ γ >0+	−193.66	−298.93	−190.39	−189.93	Elevation	
μ γ 0+	−155.03	−178.19	−158.49	−163.56	Elevation	
Temperature variability	μ γ >0+	−200.43	−235.13	−196.56	−197.42	Elevation	
μ γ 0+	−50.54	−82.55	−47.24	−46.72	Elevation	
Productivity rate	μ α >0+	−115.69	−116.98	−118.07	−131.06	Latitude	
Recruitment rate	μ β 0+	−363.13	−359.26	−369.61	−363.16	Distance	
Strength of density dependence	μ β >0+	−8.38	−8.43	−12.25	−23.19	Latitude	
Survival rate	μ δ >0+	−62.54	−62.64	−65.39	−68.88	Latitude	
Chub	Average temperature	μ γ >0+	−682.02	−800.46	−692.26	−679.58	Elevation	
μ γ 0+	−860.57	−967.08	−876.46	−858.79	Elevation	
Temperature variability	μ γ >0+	−587.89	−698.38	−584.56	−586.36	Elevation	
μ γ 0+	−361.64	−419.49	−358.73	−359.85	Elevation	
Productivity rate	μ α >0+	−278.44	−323.66	−281.23	−283.25	Elevation	
Recruitment rate	μ β 0+	−571.99	−573.06	−584.55	−597.6	Latitude	
Strength of density dependence	μ β >0+	131.5	124.65	135.05	124.34	Latitude	
Survival rate	μ δ >0+	−206.12	−214.43	−204.24	−206.63	Elevation	
Roach	Average temperature	μ γ >0+	−455.68	−558.56	−458.19	−455.13	Elevation	
μ γ 0+	−797.41	−866.15	−806.89	−795.79	Elevation	
Temperature variability	μ γ >0+	−424.75	−511.74	−421.17	−428.4	Elevation	
μ γ 0+	−755.84	−784.64	−757.28	−752.73	Elevation	
Productivity rate	μ α >0+	−357.98	−411.26	−367.68	−355.22	Elevation	
Recruitment rate	μ β 0+	−299.57	−335.09	−295.97	−299.99	Elevation	
Strength of density dependence	μ β >0+	59.26	32.43	62.13	49.66	Elevation	
Survival rate	μ δ >0+	−143.24	−149.86	−140.85	−148.36	Elevation	

Regardless of the species or the parameter considered, the null model was never ranked as the best model (Table 2). For barbel, four out of the seven factors were related to elevation, three to latitude (productivity rate, strength of density dependence and survival rate) and only one to the geographic range center (recruitment rate). For chub, six factors were related to elevation, two to latitude (strength of density dependence and recruitment rate) and zero to the geographic range center. All parameters of roach were related to elevation.

For all species and both size-classes, we always found stronger support for a differential contribution of temperature-related factors along elevation than along latitude or the distance to the geographic range center (Table 2). Furthermore, the contribution of temperature-related factors varied similarly along elevation for the three species and the two size-classes (Fig. 5). Indeed, we mostly found negative relationships between elasticities and elevation indicating (1) a rather negative effect of both factors on high elevation populations but a positive effect on low elevation populations and (2) a stronger contribution of temperature variability on high elevation populations but a stronger contribution of average temperature on low elevation populations (Fig. 5). Departures from this global pattern were nevertheless detected with e.g., an increasing and positive contribution of average temperature on the dynamic of  >0+ individuals along elevation for barbel.

Figure 5 Spatial patterns in the contribution of biotic and abiotic factors to the population dynamics of the three species as a function of the distance to the geographic range center, latitude or elevation.

Only relationships of the most supported models are displayed. In each panel, points represent elasticity values for a given population with size proportional to the corresponding absolute value and colors indicative of the sign of the value (green = positive; blue = negative). For each panel, the adjusted R2 is provided. Likelihood ratio tests were all significant (p < 0.01 in all cases). Grey zones show coefficients affecting the dynamic of 0+ individuals whereas white zones show coefficients affecting the dynamic of >0+ individuals.

More complex spatial patterns were detected when considering the contribution of biotic factors (Fig. 5). Regarding the productivity rate (μα>0+), the barbel presented a negative relationship with latitude indicating a stronger and positive contribution of this parameter to the dynamic of >0+ individuals for populations located at low latitudes whereas the opposite trend was observed for chub and roach along the elevation gradient (i.e., a higher contribution for populations located at high elevations). We also found evidence for a decrease in the contribution of the survival rate on the dynamic of >0+ individuals (μδ>0+) along elevation (for roach and chub) or latitude (for barbel). Similarly, the three species presented a positive relationship between the strength of density dependence (μβ>0+) and either latitude (barbel and chub) or elevation (roach), thus indicating a stronger negative contribution (since the parameter was negative, a positive relationship indicated a trend toward zero) of this parameter in low elevation populations. For the recruitment rate (μβ0+), while barbel presented evidence for an increasing positive contribution of this parameter as the distance to the geographic center increases, the two other species presented a negative relationship along latitude (for chub) or elevation (for roach) indicating a lower contribution of the recruitment rate on high elevation/latitude populations.

Discussion

Identifying the factors driving population dynamics is paramount if we are to effectively manage populations and prevent local extinction. To date, most studies have focused on explaining temporal variation of single population abundances (Coulson, 2001; Koons et al., 2015), hence ignoring spatial variation across populations (Frederiksen, Harris & Wanless, 2005; Pearce-Higgins et al., 2015). Here, we used length-frequency histograms and population abundance data to better understand the spatial drivers of the population dynamics of three freshwater fish species. Importantly, by integrating elasticity analyses into the modeling framework, we not only estimate the effect of biotic and temperature-related factors on fish population dynamics, but also evaluate their relative impacts (Koons et al., 2015). Despite contrasted life-histories of the three species, we found that the large-scale ecological determinants of their population dynamics were similar, with a stronger influence of biotic factors over temperature-related ones. Yet, the contribution of the factors strongly varied depending on the location of population within species’ distributional ranges and appeared to vary depending on elevation, latitude and the distance to the geographic range center.

The hierarchical structure of our modelling framework allowed us to investigate the large-scale ecological determinants of population dynamics while accounting for site-specific effects (Clark, 2005; Dorazio, 2016). Interestingly, we found that spatio-temporal variations in the abundance of the three species were driven by similar processes and were mostly influenced by factors acting on the dynamics of >0+ individuals. Specifically, >0+ abundances appeared to be strongly affected by density dependence, thus suggesting that most populations are at their carrying capacity and are regulated by competition between individuals for resources (Sæther et al., 2008). Though the effect was less clear, the dynamics of the three species also tended to be similarly affected by temperature-related factors, with both the mean and variability of temperature affecting freshwater fish population dynamics (Lawson et al., 2015). Overall, we found a positive relationship between temperature variability and the population growth rate, suggesting that conservation actions designed to buffer populations against environmental variability could in some cases reduce population growth rate (Lawson et al., 2015). The convex relationships found for the three species along the gradient of average temperature suggest that abundances are higher at both extremes of the temperature gradient. While this could be explained by local adaptations or the effect of other ecological processes (e.g., release of competition pressures due to competing species being negatively affected at both ends of the gradient), this result contrasts with both theoretical expectations (Pironon et al., 2017) and recent empirical patterns (Waldock et al., 2019) showing concave relationships between population abundance and temperature. An alternative explanation for these convex shapes could be that the sampled populations do not cover the full distributional range of the species, implying that species response curves are likely truncated (Thuiller et al., 2004). Furthermore, 95% credible intervals are rather wide suggesting some uncertainties in the estimated relationships. The similar responses observed for the three species, despite contrasting life-histories, either suggest that differences in traits are not strong enough to entail differences in population dynamics or that other factors not accounted for in this study (e.g., habitat quality, discharge) have an overwhelming influence on the observed dynamics. Such common dynamics are in line with the widespread phenomenon of population synchrony, whereby populations tend to fluctuate in a similar way in various taxonomic groups (Liebhold, Koenig & Bjørnstad, 2004) including fishes from French streams (Chevalier, Laffaille & Grenouillet, 2014; Chevalier et al., 2015).

Yet, despite obvious commonalities in large-scale drivers, specific patterns in line with species’ ecology were apparent. For instance, chub presented a higher contribution of the productivity rate to population dynamics than the two other species, suggesting that this species has a greater ability to grow from low density - and therefore to recover from disturbance (Oliver et al., 2015). This pattern matches well with the periodic strategy of chub that is favored in predictable (e.g., seasonal) and extended environments (Winemiller, 1992). Similarly, and despite the limited information contained in the data (see methods), we found a tendency toward a stronger contribution of the recruitment rate for roach and chub than for barbel, a pattern in line with the reported fecundities of the three species (Table 1; Kottelat & Freyhof, 2007). The populations dynamics of chub and roach also appeared to be more strongly influenced by the survival rate than for barbel. However, given data limitation, this results must be interpreted carefully as it may simply reflect the fact that barbel females tend to mature at a later stage (∼5 years) than the two other species (∼2–4 years; Keith et al., 2020). Overall, the general evidence we found for a larger contribution of biotic than temperature-related factors on the dynamics of both size-classes suggests that these species are rather controlled by deterministic processes which could be interpreted as evidence that they are unlikely to be strongly affected by future climate warming. However, density-dependence can also be an important mechanism in disturbed populations which can limit its ability to withstand climate change. For instance, if external factors suddenly change the availability of resources, an already disturbed population can become increasingly dependent on biotic control (either bottom up or top down) (Henley et al., 2000). The fact that we found strong spatial variations in the relative contribution of temperature-related factors indicates that environmental conditions can still have important effects on local populations. Furthermore, the overall low contribution of temperature evidenced here could be explained by a number of factors, including a poor correlation between air and water temperatures, the use of a coarse resolution not representing the conditions at the sampling sites, or potential interactions with other drivers not accounted for in this study (e.g., discharge or habitats).

Some factors displayed stronger spatial variation than others. For instance, we found larger spatial variation in the contribution of biotic factors affecting the dynamics of >0+individuals (productivity rate, strength of density dependence and apparent survival rate) than for 0+ individuals (apparent recruitment rate). The contribution of temperature variability also tended to be more spatially variable than the one of average temperature. Such spatial variations have been related to a number of geographic gradients including elevation (Dostálek, Rokaya & Münzbergová, 2018), latitude (Turchin & Hanski, 1997), species thermal maximum (Jiguet et al., 2010) or the distance to species’ range limits (Williams, Ives & Applegate, 2003). These geographic patterns suggest that intraspecific variation in population dynamic processes may be predicted from knowledge about the geographic location of populations (Sæther et al., 2008). While some species seem to conform to a given geographic gradient (e.g., elevation), results are not always consistent, with other species showing no patterns and sometimes even opposite patterns (Dallas, Decker & Hastings, 2017; Santini et al., 2019). These inconsistencies do not necessarily mean that there is no pattern, but perhaps that the geographic gradient considered is not appropriate. We here sought for evidence of two common geographic patterns: one related to elevation or latitude and the other related to the distance to the geographic range center. These two gradients have both theoretical and empirical underpinnings (Turchin & Hanski, 1997; Sæther et al., 2008; Pironon et al., 2017; Yañez Arenas et al., 2020) but were never confronted to date.

For the three species we considered, most of the factors contributing to population dynamics showed stronger empirical support for an elevational rather than for a distance-based or latitudinal gradient. This was particularly the case regarding temperature-related factors. Nonetheless, the spatial pattern was not always consistent with the range shift hypothesis (Comte & Grenouillet, 2013; Comte et al., 2020). For instance, while an upward shift has been documented for the three species at the trailing edge (meaning extirpations; Table 1), we estimated a rather positive effect of both temperature-related factors on low-elevation populations. Discrepancies were also evident at the leading edge for chub and barbel. For instance, the leading edge of chub was predicted to shift upward whereas we estimated a negative effect of both temperature-related factors on high-elevation populations. The opposite pattern was observed for barbel, i.e., estimated downward shift but positive effect of average temperature. While these discrepancies can be explained by different factors, including abiotic factors not accounted for (e.g., water quality—Britton, Davies & Pegg, 2013; river flow—Bergerot & Cattanéo, 2017), extinction debts at the trailing edge (Kuussaari et al., 2009) or colonization credits at the leading edge (Rumpf et al., 2019), the dendritic structure and the flow directionality of the river network that strongly constrains population dynamics (Larsen et al., 2021), the way range shifts were estimated (Comte & Grenouillet, 2013) and how population dynamics were modelled (this study); they can also be explained by the fact that temperature-related factors contributed little to the population dynamics of the three species. The fact that the contribution of biotic and temperature-related factors varies differently along geographic gradients can explain why many studies failed to find general evidence for abundance-distribution patterns in natural settings (Dallas, Decker & Hastings, 2017; Santini et al., 2019). Similar to our results, other studies have shown that different demographic parameters can present opposite geographic trends, through different responses along environmental gradients; a process known as demographic compensation (Csergő et al., 2017). Altogether, these results call for more detailed approaches (i.e., considering different components of population dynamics along with their contribution) if we are to better understand the spatial drivers of population dynamics and the associated species range dynamic at large spatial scales. Moreover, although we highlighted spatial gradients, the spatial factors we have considered are “indirect predictors” in the sense that they do not affect species directly. They are however useful because they usually correlate well with other important variables that have a direct influence on species’ physiology (e.g., temperature) and which are of great interest to managers and stakeholders (Guisan & Zimmermann, 2000). Therefore, instead of searching for general spatial patterns, future studies should seek at identifying what are the proximal determinants of spatial variations in population dynamics. Such factors have already been identified for some species. For instance, water quality (Britton, Davies & Pegg, 2013) and habitat availability (Farò, Zolezzi & Wolter, 2021) have been shown to be important determinants of spatial variations in barbel’s population growth rate. In marine fishes, Wang, Kuo & Hsieh (2020) have recently shown that truncated age structure, warming and spatially heterogeneous temperatures can enhance population spatial variability. Regarding our three studied species, population dynamics mainly varied along elevation which is strongly correlated with a number of direct predictors including temperature, discharge, depth or particle size of sediment.

This study can be expanded in different ways to improve our understanding of spatial variation in population dynamics. First, our data only allowed us to discretize two size classes, thus limiting our ability to go deeper into the demographic mechanisms underpinning population dynamics (Bergerot & Cattanéo, 2017). For instance, the recruitment rate can be influenced by many factors which are not considered here, such as age, maturity stage, sex ratios and size that can have strong influence on the fecundity of spawning individuals and the associated recruitment success (Vilizzi, Copp & Britton, 2013). Stage structured population models (Caswell, 2001) and integral projection models (Merow et al., 2014) could be used to gain further insights about the drivers of population dynamics, e.g., by drawing inferences on survival probabilities or reproductive rates for different age classes or individual states. These models are however extremely data-demanding which may preclude their application at large spatial scales, though large-scale databases are being developed (http://demography-portal.ex.ac.uk/). Despite providing limited information on demographic parameters, the biodiversity monitoring data and the analysis workflow we have used made it possible to gain additional insights on population dynamics, compared to traditional models that only focus on population abundance without accounting for size-specific differences. In this sense, our modelling framework can be seen as something in between traditional population dynamic models (Keith et al., 2008) and matrix population models (Caswell, 2001). Second, our modelling framework could be improved in different ways. For instance, depending on the data at hand (e.g., spatial or temporal replicate), one can include a state-space or N-mixture component to the Bayesian population dynamic model to account for spatial (and temporal) variation in detection probability (Zipkin et al., 2014). Similarly, one can imagine computing parameter elasticities and testing the effect of spatial covariates directly within the Bayesian model to allow uncertainty to fully propagate along the different layers of the model. We here chose not to use this approach because we built on previous studies that used a similar approach as ours (Grenouillet et al., 2001; Koons et al., 2015), but also because such modifications would lead to a very complicated model with an associated lower interoperability and a higher computation time. Third, we only considered the effect of temperature but future studies should also consider other potentially important variables (e.g., hydrology, habitat diversity) to evaluate the cumulative contribution of different abiotic factors to population dynamics. Considering other abiotic factors would also make it possible to account for synergistic effects that may sometimes affect populations in unexpected ways (Morrongiello et al., 2021). Such effects could for instance help explain the apparent contrast between a leading edge for chub predicted to shift upward but a negative effect of temperature on high-elevation populations (e.g., species running out of optimal niche and slowly forced to extend in potentially sub adequate environments). Fourth, the abundance data we had covered a much wider gradient for elevation than for the two other spatial covariates, particularly concerning the distance to the geographic range center (Fig. 4). If our populations had been more evenly distributed across the whole range, perhaps the distance to the geographic range center or the position along the latitudinal gradient would have had stronger effects.

Conclusions

Illuminating the factors driving spatial variations in population dynamics is a long-standing goal in Ecology (Brown, 1984; Sæther et al., 2008). Here, we evaluated whether spatial variations in the contribution of different factors affecting the population dynamics of three freshwater fish species could be explained by two commonly observed biogeographical patterns: one related to elevation or latitude (range-shift hypothesis) and the other related to the distance to the species geographical range (abundant-center hypothesis). We found that both, biotic and abiotic factors rather tended to vary along elevation or latitude providing stronger support for the range-shift hypothesis. Yet, the contrasted spatial patterns highlighted here suggest that observed spatial variations in population dynamics can be the result of different processes acting in opposite ways with e.g., one process (e.g., climate change) driving spatial variation in one way and another process (e.g., competition for resources) driving spatial variation in another way. This dual influence may partly explain why some recent studies found low evidence for geographic gradients (Dallas, Decker & Hastings, 2017). Overall, our results call for more detailed approaches (considering different demographic parameters along with their contribution) if we are to better understand the drivers of spatial variation in population dynamics and associated abundance-distribution patterns.

Supplemental Information

Supplemental Information 1 Summary statistics and graphics regarding raw data

Click here for additional data file.

Supplemental Information 2 JAGS model code

Click here for additional data file.

Supplemental Information 3 How the JAGS data can be uploaded in R and what are the variables contained in each file

Click here for additional data file.

Supplemental Information 4 Data to feed the JAGS model for the Barbel

Click here for additional data file.

Supplemental Information 5 Data to feed the JAGS model for the Chub

Click here for additional data file.

Supplemental Information 6 Data to feed the JAGS model for the Roach

Click here for additional data file.

We are indebted to the French Biodiversity Agency (OFB) for providing fish data.

Additional Information and Declarations

Competing Interests

Author Contributions

Data Availability

The authors declare there are no competing interests.

Mathieu Chevalier conceived and designed the experiments, performed the experiments, analyzed the data, prepared figures and/or tables, authored or reviewed drafts of the paper, and approved the final draft.

Pablo Tedesco analyzed the data, authored or reviewed drafts of the paper, and approved the final draft.

Gael Grenouillet conceived and designed the experiments, authored or reviewed drafts of the paper, and approved the final draft.

The following information was supplied regarding data availability:

All the data used to fit the JAGS models for the three freshwater fish species are available in the Supplementary Files.

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
