# Peer review of "Spatial patterns in the contribution of biotic and abiotic factors to the population dynamics of three freshwater fish species"

_PeerJ, doi:10.7717/peerj.12857_

## Round 0.1 · original submission · Major Revisions

The recommendations of three reviewers differ, but all have provided extensive comments on the manuscript. Given their concerns about the need to provide clear justification and explanation for the hypotheses being tested, the lack of clarity regarding availability of data and other materials (e.g. code associated with the analyses), and the need to provide clearer explanations of methods and results I would request that this manuscript be significantly revised following reviewers; comments prior to resubmission. As one of the reviewers rejected the paper outright, I would ask you to address all of their concerns explicitly in your revisions letter.

·

Basic reporting

The manuscript is adequately presented and clearly structured. While the methods section may be a bit long and on the more verbose side, it seems in this case justified given the complex statistical and modelling approach. No issues with language outside of small comments in passing. The figures and tables are also clear and nicely presented, although some edits are probably needed for figure 1 (see General Comments). Raw data is available for checking and is easily comprehensible.

Experimental design

I first commend the authors for the thorough and extensive analysis they conducted for this manuscript. Not only does it utilize an impressive dataset derived from a monitoring program spanning a large area, but it does so in a very clear and advanced modelling framework. The research question and framing of the introduction is well-crafted and perfectly sums up the background, theory and expected contribution behind this analysis, particularly the articulation of the three study species’ in relation to the abundant-centre / range-shifts hypotheses.

As written previously, the methods section is a bit long in comparison with the rest, but this is often warranted when complex modelling is involved, and it has here the merit of being clear and well presented. I am not especially well-experienced with Bayesian frameworks and I therefore won’t dive too much into the nitty-gritty of the statistics (although I would certainly be interested to read the comments of other reviewers), but the modelling approach itself was fitting and nicely presented. I am particularly appreciative of the elasticity analysis and its incorporation into testing the two biogeographic hypotheses. This brings a new and welcome layer of analysis as well as more caution to the discussion, which is something sadly too often left out from modelling studies.

Validity of the findings

The respective influences of biotic and abiotic factors on population dynamics is a widely studied (and debated) topic, but multi-populations or species comparisons integrating spatial elements are less commonly seen. Here, the authors use data from a long-term freshwater monitoring program to develop population dynamics modelling for three fish species with varying life histories and environmental preferences to quantify the relative importance of biotic and abiotic factors. In addition, the results are further used to evaluate spatial patterns in fish response to each factor following two (non-exclusive) ecological hypotheses. Studies on species distributional changes under shifting environmental conditions have been a key aspect of modern ecology, but freshwater ecosystems have been notoriously understudied and neglected in comparison to their marine counterparts. As such, this manuscript provides a welcome addition to a growing body of literature and provides interesting new perspectives on the influence of biotic / abiotic factors on freshwater fish dynamics, as well as encouraging further studies.

The discussion did a great job to provide a summary of the results and re-contextualize them with the research questions. A clear and concise conclusion is also provided at the end to summarize the key findings, encompassing both the modelling results on the factors driving population dynamics, and how they relate to the two mechanisms suspected to drive species distribution. Overall, this is a high-quality work.
I’d however be careful with some of the interpretations in the discussion. For example, the larger effects of density-dependent factors are here defined as a sign of stable populations near carrying capacity, and the authors therefore present several elements pointing toward a potential “resilience” of these fish populations to climate change. While it is true that density-dependence is often a significant control mechanism seen in larger and healthier populations, there is a growing body of evidence showing strong density-dependent effects appearing in disturbed populations past a certain threshold, especially when environmental changes are involved. Indeed, if external factors suddenly change the availability of resources (for example a long-term warming leading to a gradual mismatch between prey / predator peak recruitment season) an already disturbed population can become increasingly dependent on biotic control (either bottom up or top down) where the density-dependent effects will then limit its adaptability to further change.

In the manuscript there is a paragraph briefly touching on the potential factors not considered here behind the opposed effects of biotic / abiotic factors for some populations (lines 389 – 396), but I encourage the authors to be a bit more cautious on some of their conclusions as more and more evidence shows that these different factors often act synergistically in unexpected ways (see for example Morrongiello et al 2019, Synergistic effects of harvest and climate drive synchronous somatic growth within key New Zealand fisheries). For example, here the apparent contrast between a leading edge for chub predicted to shift upwards but a negative effect of climate change on high-latitude populations could very much indicate that the species is running out of optimal niche and is slowly forced to extend in potentially sub adequate environments.

Additional comments

Not an issue per se, but at times I got confused with the presentation of one of the two hypotheses (range-shift). While the results and discussion make it clearer further down the manuscript, the initial presentation of the range-shift hypothesis gave me the wrong impression that this study would be looking at latitudinal changes, i.e. poleward movement of freshwater species with abundances in the southern sites decreasing and those in northern sites increasing. This conflicted with the repeated use of “altitude” but because it wasn’t clearly stated that the range-shift hypothesis was here tested for altitudinal changes, it took me a re-read to correctly understand (though some may argue that these two are here not mutually exclusive). I think an extra sentence in passing in the methods could help to clarify.

Figure 1
In the caption, the word “raw” was wrongly used in place of the word “row”, this needs to be corrected. I think it would also be easier to read with annotated panels (say A to H). The panels are also rather small and difficult to read, in particular the smaller plots above. I’m not sure if the figure should be reorganized more vertically (3 rows instead of 2); be flipped in the published version; or if these elements should be made bigger.

Figure 4
Perhaps a minor comment, but consider changing the point colour palette from green/orange to something more contrasted like green/blue for those with colour eye deficiencies like colour-blindness.

Reviewer 2 ·

Basic reporting

In this study, the authors use statistical models to explore the influences of biotic and abiotic variables on the population dynamics of two age classes of three fish species with contrasting life histories across part of their geographical range - France. They followed this exploration with more statistical models to test whether experimental variation in these biotic and abiotic variables were best described by "range-shift" or "abundant-centre" spatial patterns. Aside from their main explorations, the authors also describe statistical models used to allocate individual fish of each species to one of two age classes (0+ and >0+) based upon their length.

Overall, I found the study interesting, well-written and well-structured. Nevertheless, I also found it difficult to understand. In part my difficulties arise from incomplete descriptions of the methods and theory. Specifically, the overall question(s) and hypothesis(es) or prediction(s) are not clearly stated, with the result that (in my opinion) the study seems to meander towards the conclusions. The question presented in the last paragraph of the Introduction is not clear: it does not mention the fact that the species had contrasting life-history strategies, nor that the study of spatial patterns was actually a test between competing a prior hypotheses (no effect versus the range-shift and abundant-centre hypotheses). In my opinion, these omissions suggest that the comparisons between species and spatial patterns were not central to the study design, but rather included as an afterthought. If I am wrong and they were central to the study design, then one could imagine an analysis that included model terms representing those effects in the population dynamical models and which could be used to test their importance, e.g., using model comparison (as is done in the current spatial patterns analysis). Specifically, “species” could be added as a three-level factor and coefficients representing altitude/latitude and distance from range centroid could be included in each age-class-specific model. I understand that this would differ from the current approach that examines how variations in each parameter due to the elasticity analyses are associated with each spatial pattern, but I remain unconvinced that this approach is any more interesting that the approach I suggest...

Experimental design

Again, I found the study to be interesting, but not entirely convincing. This is because it did not seem to address clear question(s) and hypothesis(es) or prediction(s).

I could not find it written whether the population dynamical models were fit for all species combined or separately (I assume that they are fit for species separately). Figure 1 suggests that the responses between the different species were quite similar. Would it be interesting to test for species differences in their responses to these biotic and abiotic variables thereby testing whether their "contrasting" life-histories are important in this study?

The “range-shift” hypothesis is not limited to vertical range-shifting (altitude) as described in this study, but also includes horizontal range-shifting (latitude) (e.g., Parmesan, C., Ryrholm, N., Stefanescu, C., Hill, J. K., Thomas, C. D., Descimon, H., ... & Warren, M. (1999). Poleward shifts in geographical ranges of butterfly species associated with regional warming. Nature, 399, 579-583). I therefore question the results of the spatial exploration - would the results be similar if latitude was tested instead of altitude? Personally, I don’t think that Figure 3 brings sufficient information to the study to merit its inclusion, particularly because the interpolated values are shown on different scales for all maps. In my opinion, Figure 4 ought to include plots where the null model was preferred, but that no trend line should be fitted.

The description of the methods used in the “Spatial pattern in the contribution of
biotic and abiotic factors” section were too sparse for me to understand what was done (although it seems that the elasticity values were used as the response variables; lines 246-248). Personally, I am not entirely convinced by this analysis and I would prefer to see a combination of model fitting and model selection used when fitting the population dynamical models to test the support for the competing a prior spatial pattern hypotheses.

I did not well follow the descriptions of the mixture models used to separate age 0+ and >0+ individuals, perhaps because the procedure is quite involved? I do understand random forests, but I could not fully understand what was done prior to the random forest fitting.

I could not find accessible geographical range polygons for the three species on the IUCN Red List. I would appreciate more information about how distance to the geographical range centroid was calculated.

While the functions included to account for biotic effects, such as apparent recruitment rate and density-dependence, were somewhat crude (e.g., are all >0+ individuals mature and successful spawning females?), I commend the authors for acknowledging and attempting to account for such nuanced effects in their analyses.

Validity of the findings

I also struggled to understand the study because the raw data and analysis code were not described or provided. I note that the authors sent the raw data with their submission, but I also note that the population abundances were provided as densities rather than counts, there is no length data to repeat the age classification, and neither is there a description of the data provided. Perhaps worse than those, it appears from examination of the data provided that there were many zero counts, particularly among age 0+ fish, but that this is not discussed or shown anywhere in the study write-up. Personally, I believe strongly that the raw data should be shown and if space is an issue, then Figure 3 could be dropped. Aside from the data, I note that the authors state that they provide the analysis code, but I did not find it. Although perhaps not a requirement to understand the paper (provided the questions, predictions and Materials and Methods are better described), providing the code would make it easier to understand the study.

Additional comments

No additional comments.

Reviewer 3 ·

Basic reporting

- I think the introduction section reads well and is well referenced.
- L55-75. This section would benefit from the addition of the author’s hypotheses regarding the variables tested for each age-class and species – which is somewhat provided in the first section of the methods, however I would like to see more specific hypotheses supported by literature on the ecology of the fish species studied, for example, any existing evidence on relationships between temperature and population vital rates? I would also like more specification of the direction of effects, e.g. L95-96 “(either positively or negatively)” – which direction for each species? The detail on L58-60 could be omitted as this information is already supplied in the methods.
- I would like to see a plot or some summary information of the temperature data to see how much the annual mean temperature variables vary between sites. Also, this would show whether mean temperature ever exceeded the species maximum values as shown in Table 1. And whether a quadratic effect was necessary? Perhaps the authors can provide their hypotheses for this effect.
- I would recommend a visualisation of the species range and geographic centre of the range, considering the spatial element is the focus of this study. Also, the table suggests there’s quite a difference in scale, with roach having a much larger range relative to the other species. And where the leading/trailing edges are, etc. This figure could be combined with topography to visualise the differences in altitude across the study sites.
- More of the Discussion section should focus on placing the findings of the study in the context of the existing knowledge of the ecology and population dynamics of each fish species. For example, comparing the literature of temperature effects/density-dependence on population abundance, recruitment rate, survival, etc of each species and especially any literature considering spatial variation in population response of these species to intrinsic and extrinsic factors (e.g. Britton et al., 2013 DOI:10.1111/j.1600-0633.2012.00588.x; Faro et al., 2021 DOI: 10.1016/j.jenvman.2021.112100)
- L219. There is no Appendix S1 and thus, no JAGS code – this would be a valuable addition, which I commend the authors for preparing and perhaps has been omitted by accident?
- Fig 2. I appreciate the consistency of using terminology from the equations in the figure legends, but for ease of understanding, I would advise simpler labelling, e.g. “Mean temperature (linear/quadratic)”, “Recruitment rate”, etc. Also, in the figure legend ‘the first raw’ = ‘the first row’?
- Fig 3. The figures should be on the same colour scale to aid interpretation, especially as the maps and legend values are too small to quickly glance and read, also it is a bit misleading when the maps are similar colours but the scale is a different range.
- L295-296: ‘size-classes’ be consistent with terminology, these were first described on L59 as age-classes
- I think the term ‘abiotic factors’ is unnecessary throughout the manuscript – the authors only looked at temperature so would it be clearer to refer to these as temperature variables. Also, sometimes these variables are referred to as ‘climatic variables/factors’ e.g. L265 – again, consistency with terminology helps the reader to follow the study.
- L333. I recommend changing ‘growth rate of >0+ individuals‘ to ‘population growth rate’ as it currently reads as the somatic growth rate of individuals
- Table 1 should be referenced.
- L172. I appreciate the inclusion of model equations – it really helps the reader understand the analysis. Do the authors mean to include a term to show that models were built separately for each species? Or could add this information to the text to avoid repetition.

Raw data
- From my understanding, the authors supply the following raw data for each species: Densities of 0+ and >0+ for each year and site, length of time-series, mean temperature, variation in temperature, the number of time-series, but I am not sure what $samp is? Perhaps the site area? It would be helpful to supply more descriptive identifiers either in the dataset itself or supporting text file.
- The fish data supplied are densities, but the models are fit to count abundance data – I understand the authors include log(site area) as an offset in the models, but with these data, the analyses aren’t reproducible, and it would be beneficial to see the raw data.
- No length data are provided, which are used to age the fish. I would like to see a summary plot of the length-frequency histograms used to age the fish for each species in the supplementary material.
- No data are supplied for the altitude and distance to geographical centre of distribution, which are essential for the third part of the analysis.
- A general comment is that it would be beneficial to visualise the raw data or a summary of the raw data. I think the supplementary material are a suitable place for this, or particularly the main manuscript when describing response variables and explanatory variables used in the analysis.

Experimental design

- In my opinion, not accounting for a likely effect of density-dependence on 0+ abundance is a significant omission, as in many freshwater fish species, this is the age where density-dependent processes are considered to have the strongest effect on population sizes (e.g. Grenouillet et al., 2001 Freshwater Biology; Jonsson et al. 2001 Journal of Animal Ecology DOI: 10.1046/j.1365-2656.1998.00237.x). Furthermore, including the abundance of >0+ to estimate recruitment rate is too crude: many factors which are not considered here, such as age, maturity stage, sex ratios, size and thus fecundity of spawning individuals, will influence recruitment and should be accounted for. Also, the authors assume constant survival of recruits independent of recruit density. To improve this parameter, I recommend the authors to explore the use of a recruitment function, such as Ricker or Beverton-Holt, where recruitment from eggs to 0+ is estimated as a density-dependent function (and can be modified by explanatory variables as required).
- I find the elasticity analysis and the spatial pattern analysis to be difficult to understand and reconcile with the research question. I do not understand why total population abundance (the sum of 0+ and >0+ abundances) was used to test the influence of a partialized effect of each explanatory variable. Different variables were included in models of each age-class so it does not seem appropriate to test for the relative influence of all variables (which were chosen specifically for the particular age-class) on the total population abundance. And of the variables that were included in both models, some influenced 0+ and >0+ abundance differently, for example temperature variation appears to influence 0+ abundance in all species linearly, but >0+ abundance in all species in a quadratic manner, therefore it does not seem appropriate to combine the abundance of each age-class to determine relative influence of each variable. I would recommend that either the elasticity analysis or another Bayesian variable selection technique (see Hooten and Hobbs, 2012 DOI: 10.1890/14-0661.1
for a review of techniques) is performed for each age-class model separately. I also find the step of first generalising the influence of explanatory variables on age-specific abundance across spatial scales by including site as a random effect, and then attempting to describe these patterns by spatial covariates to be confusing. Would it not be better to test for spatial effects within the initial abundance model?
- L115-117. I note in the methods that the authors took steps to discard time-series data that contained at least 50% of non-null captures for at least 15 years. Was this to maximise the amount of catch data >0 in the time-series? From looking at the raw data, when these data are split by age-classes, there seem to be quite a lot of time-series that include >50% non-null captures, especially in the 0+ age-class (Barbel >0+: n = 7, 0+: n = 21; Chub >0+: n = 2, 0+: n = 55; Roach >0+: 5, 0+: n = 61). Did the authors investigate whether these data are zero-inflated and do they need to consider the amount of zero captures in the model structure? It would also be useful to note the distribution of these sites with high levels of zero-captures to ensure that the representation of sites with limited >0 catch data is not geographically bias and therefore suitable for spatial analysis. For example, high numbers of zero captures in the 0+ compared with low numbers of zero captures in the >0+ could suggest that the sites surveyed are not appropriate juvenile habitats. This could be a more reasonable explanation than consecutive years of failed recruitment or poor sampling of juveniles. It also might suggest that these data are not suitable to investigate influences of 0+ population abundance.
- L193-194. The apparent survival transition – is it possible that fish in the >0+ age-class were different ages? If so, estimating the probability of surviving between subsequent age-classes (i.e. 0+ to 1+) would not be suitable. If all of the fish are 1+ in the >0+ age-class, perhaps this terminology would be clearer?
- L112. Electrofishing surveys taken in periods of low flow? Could the authors clarify what ‘low flow’ represents? How might this influence the population dynamics sampled, i.e. abundance and size of individuals, and the subsequent dynamics investigated in this study? (e.g. Allouche and Gaudin, 2003. Doi: 10.1034/j.1600-0706.2001.940310.x; Britton and Pegg 2011 Doi: 10.1080/10641262.2011.599886).
- L115-117. It would be useful to know the sampling protocol that the authors chose as the reference that is cited (Poulet et al., 2011), state different methods: 2-pass depletion fishing, point sampling and fractional sampling. Depending on the efficiency of the sampling, the authors could attempt to account for imperfect sampling by estimating capture probability for each class in an age-structured state space model (Parent & Rivot, 2003. Introduction to hierarchical Bayesian modelling for ecological data) and estimating a ‘true abundance’ with which to describe with explanatory variables.
- Do the authors think that their temperature variables captured changes that would be relevant to preferences/behaviours described in the species life histories, i.e. that opportunistic strategy maximises the colonizing capability of species in environments with frequent changes at small temporal and spatial scales, or periodic strategy favoured in seasonal environments? It seems that a mean annual temperature variable might be too coarse to distinguish short-term or seasonal fluctuations in temperature.
- Why did the authors only choose to include temperature as an abiotic variable and not some discharge related variable? I appreciate that with the large number of study sites, it would be difficult to obtain data for other environmental and habitat variables. Could the authors provide more justification for their choice in the methods.
- L98-106. Are the leading and trailing edges of the population ranges always high and low-altitude populations, respectively? I would expect a latitudinal shift also as populations migrate to cooler, higher latitudes. There is existing evidence for an effect of temperature and latitude on life history traits of barbel and roach (Britton et al., 2013 DOI:10.1111/j.1600-0633.2012.00588.x). Why did the authors not consider latitude as an explanatory variable to test for evidence of the range-shift hypothesis?
- How is the spatial distribution considered, i.e. what about sites that are closer together? Does the random effect account for this? Could the authors provide more information as to how they accounted for any spatial correlation between sites?
- L240-248. This section is not written well enough to understand the analysis done; the response variable is not stated clearly, nor the model type or distribution or whether separate models were fitted for each species. All the other models have associated equations to aid understanding, this should be replicated here.
- L181-182. Length-frequency histograms do not provide enough information to inform recruitment rate, for example male barbel tend to mature at a smaller size than females (Vilizzi et al., 2013 DOI: 10.1051/kmae/2013054)

Validity of the findings

- Fig 1. Quite similar responses of the species to the abiotic and biotic variables! This is an interesting finding and, in my opinion, should be discussed more in the Discussion section.
- L264-265. “All species presented a tendency for an effect of climatic factors on the dynamic of both 0+ and >0+ individuals…” Can the authors provide more description of the temperature effects – were they linear/quadratic for which species, direction of effect etc.
- L270-272. The authors state that >0+ individuals are more sensitive to changes in biotic factors than 0+ individuals, but they omit testing the effect of density-dependence on 0+, which is considered a strong regulator of population size in juvenile freshwater fish, and thus, I don’t believe they can draw this conclusion from this analysis.
- L275-278. The lack of relative contribution of abiotic variables could indicate that the temperature variables used, i.e. mean annual temperature was not at suitable resolution to capture an effect, or that other abiotic factors might be more useful in describing abundance, such as discharge, habitat and prey availability, etc. The authors do note that other variables could be important (L424-425), but could they expand on potential limitations of the temperature variables used?
- L284-285. “For most factors we found that these spatial variations were mostly related to altitude and less so to the distance to the geographic range center”. To assess the validity of tests of relative influence of explanatory variables, it is necessary to include some information of the data used to calculate each variable or summarise the variables. Could this finding be an artifact of there being more variation in altitude rather than distance to the geographic range centre? According to the IUCN Red List maps cited as the source of species ranges, the centre of each species’ range appears to be in central Europe, so all the populations studied here will be relatively far from the centre? Again, some summary of these covariates would be useful. From Fig. 4, it looks like few values of high-altitude sites might have strong influence on the shape of the slope?
- L279. I don’t see from Fig. 2 that the effects of abiotic variables ‘outcompete’ those of biotic factors? All of the parameter estimates for the temperature variables overlap zero and thus don’t have strong effects?
- L338-339 – why would higher abundances result in a ‘release from competitive pressures’?
- L353-355. The findings of this study do not support this statement, as such it should be removed.

Additional comments

This study aims to identify the relative influence of abiotic and biotic variables on the population abundance of two age-classes of roach, chub and barbel at multiple sites throughout France. The authors use a series of analytical steps to first describe age-specific abundance as a function of the abiotic and biotic variables, then test the relative strength of each variable on total abundance by partializing out the effect of each variable. Then, finally, they attempt to identify whether there are spatial patterns in the relative strength of each variable on total abundance, by testing the influence of altitude and distance to geographic range centre and performing model selection. I think the authors attempt to answer an interesting research question and that the study benefits from a large (both temporally and spatially) dataset. However, I have numerous concerns with this study, most imperative is the statistical analyses, which ultimately, I don’t believe enable them to answer their research questions. I have provided more detail in my comments, which I hope are clear and helpful.

---

## Round 0.2 · Minor Revisions

While you have made significant improvements and additions to the manuscript, the reviewer notes that there are still some minor inconsistencies which need to be addressed. The reviewer also continues to be concerned about the assumptions made in the paper, and I would also recommend that you perhaps acknowledge these in the manuscript.

Reviewer 3 ·

Basic reporting

- Regarding a suggestion that more of the Discussion section should focus on placing the findings of the study in the context of existing knowledge of the ecology and population dynamics of each fish species, which the authors marked as ‘Done’. However, I could not find where this addition is?
- Figure 1 is a nice addition to help the reader understand the data and modelling process. The new Figure 3 and updated Figure 4 look like good additions and Figure 4 appears to be clearer to read in terms of the colour scheme, however the resolution is too low so the figures are unreadable in their current format, so I couldn’t evaluate these (the resolutions of Figs 2, 3 and 5 should also be improved as they are also difficult to read when zoomed in).
- I was unable to load in the revised .RData files to review, I received the following error message: ‘bad restore file magic number (file may be corrupted) -- no data loaded’
- L218-219; 301-302 (and please check throughout) – terminology of variables have been changed elsewhere but a few places remain with the original terms.

Experimental design

- Regards using a ‘recruitment rate’ to describe 0+ abundance: I had expected to see a more explicit parameter to account for density-dependence in 0+ abundance, such as egg biomass, or spawner biomass as a proxy for this (I stated this in my previous review). In their response to my latest comments, the authors have stated that they do not have egg data, and it seems clear that they do not have information on age-structure or maturity in the older population (or if they do, then they haven’t used it). In my opinion, the authors should clarify this situation in the Methods section (alongside the description of the analyses), in order to justify how they have attempted to account for density-dependence in 0+ abundance. Specifically, I suggest the authors add a paragraph in the Methods stating why and how it is important to account for density-dependent effects, with referenced examples for this study species, together with an description of the data available and unavailable, e.g., egg data, to account for its effect, and what assumptions had to be made in lieu of egg data. A more involved, but more convincing, addition would be to undertake a sensitivity analysis to explore the sensitivity of the results to different assumptions about the value and variation in recruitment rate.
- Furthermore, while I appreciate the sentence added to the Discussion on L584-600 describing how the study can be expanded by exploring the influence of other factors on recruitment rate, I do not consider this a substitute for the aforementioned paragraph in the Methods section (alongside the description of the analyses) stating the data limitations and attendant assumptions. In summary, I too appreciate that there has been an attempt to account for these effects, but I think the interpretation of results should be more cautious and the explanation of this method be more explicit to allow a reader to interpret the validity of this result.

- Regarding a previous comment on the apparent survival transition from 0+ to >0+: the authors have confirmed that the >0+ size-class comprises individuals of various ages, and so perpetuates my concern that survival transitions between all ages >0+ are assumed to be the same, i.e., whether they are estimating survival between 0 to 1+, or 0 to 2+, 0 to 3+ etc. This is a strong assumption. A more relaxed assumption might be to allow survival probabilities to change with ontogeny. Also, the authors describe in their responses that ‘the 0+ size class in a given year cannot be influenced by the number of recruits in the previous year because those individuals are transitioning to the >0+ size class between consecutive years’; but a 0+ fish cannot transition directly to a 2+ or older fish. Even assuming a constant survival rate between different age-classes, the survival rate should be applied sequentially. I assume and understand that the authors do not have the ages for the population, however the authors could limit the >0+ size-class to only 1+ individuals using the existing approach used to identify two size-classes using length-frequency histograms.
- Furthermore, the authors could better justify the apparent survival transition by providing some information about the species ecology. For example, average/range of ages that the species mature, any information about sex ratios in the population. This would help the reader assess the validity of this approach, such as, if fish of these species are capable of spawning at ages 1+ and older, then it would be more reasonable to assume that individuals of this age, i.e., that constitute the >0+ abundance, can contribute to the apparent recruitment rate. If the authors stick with their existing approach of two size-classes, then I would expect some comment on the limitations of this study when trying to account for apparent survival transition.

- L622-624. This addition based on previous comments regards the study findings that spatial variations were mostly related to altitude/elevation rather than to the distance from the centre of the species range might be misunderstanding my original comment: the imperative point here, is that the data points were not widely distributed throughout the species entire ranges, rather than evenly distributed, and thus the dataset seems somewhat inappropriate to test for hypotheses related to distance to centre of the species range as all the populations included in the study are in a relatively similar location in relation to the centre. This is reiterating a previous point but I feel an important one given the intentions of the study. In my opinion, I would focus more on discussing the more convincing aspects of the study such as the effects of covariates on the abundance of fish species studied and spatial variation in these effects.

Validity of the findings

no comment

Additional comments

The authors have done a large amount of work to respond to all the reviewers’ comments and I believe that the manuscript is much improved as a result. Some of my previous concerns regarding the abundance estimation and distribution of datapoints remain, and I have a few minor comments; please see details below.

---

## Round 0.3 · accepted · Accept

Thanks for responding quickly to these minor revisions - I am happy that you have addressed all reviewers' comments to the best of your ability.